# The Effect of Monoculture, Crop Rotation Combinations, and Continuous Bare Fallow on Soil CO_2_ Emissions, Earthworms, and Productivity of Winter Rye after a 50-Year Period

**DOI:** 10.3390/plants11030431

**Published:** 2022-02-04

**Authors:** Vaclovas Bogužas, Lina Skinulienė, Lina Marija Butkevičienė, Vaida Steponavičienė, Ernestas Petrauskas, Nijolė Maršalkienė

**Affiliations:** Agroecosystems and Soil Sciences Department, Agronomy Faculty, Vytautas Magnus University, K. Donelaičio Str. 58, 44248 Kaunas, Lithuania; vaclovas.boguzas@vdu.lt (V.B.); lina.butkeviciene@vdu.lt (L.M.B.); vaida.steponaviciene@vdu.lt (V.S.); ernestas@agronom.lt (E.P.); nijole.marsalkiene@vdu.lt (N.M.)

**Keywords:** continuous bare fallow, rye, monoculture, 50-year period crop rotation, cover crop green manure, pre-crop effect

## Abstract

One of the main goals of the 21st century’s developing society is to produce the necessary amount of food while protecting the environment. Globally, particularly in Lithuania and other northern regions with similar climatic and soil conditions, there is a lack of data on the long-term effects of crop rotation under the current conditions of intensive farming and climate change. It has long been recognized that monocultures cause soil degradation compared to crop rotation. Research hypothesis: the long-term implementation of crop rotation makes a positive influence on the soil environment. The aim of our investigation was to compare the effects of a 50-year-long application of different crop rotations and monocultures on soil CO_2_ emissions, earthworms, and productivity of winter rye. Long-term stationary field experiments were established in 1966 at Vytautas Magnus University Experimental Station (54°53′ N, 23°50′ E). The study was conducted using intensive field rotation with row crops, green manure crop rotations, three-course rotation, and rye monoculture. Pre-crop had the largest impact on soil CO_2_ emissions, and more intensive soil CO_2_ emissions occurred at the beginning of winter rye growing season. Rye appeared not to be demanding in terms of pre-crops. However, its productivity decreased when grown in monoculture, and the optimal mineral fertilization remained lower than with crop rotation, but productivity remained stable.

## 1. Introduction

The growing global population is driving the intensification of agricultural production [1], farmlands dominate 38% of the global land surface, and almost 30% of the global net primary production is for human use [2]. The evidence of climate change is undeniable, and it is now recognized that climate change needs to be mitigated [3]. The management of agroecosystems plays a crucial role in the global carbon cycle [4]. A continuous simplified crop rotation with the same crops decreases the potential for use of both soil and ecosystems, the negative consequences of which could be avoided through more diverse land utilization, by both sowing and growing different crops and applying different forms of tillage [5]. Agricultural systems that increase the reliance on biodiversity can reduce the risks from climate change challenges and should be considered essential to meet the great challenge of climate change mitigation [6].

The cultivation of winter rye differs from that of other cereals, and this crop affects the soil properties and soil biodiversity in a peculiar way [7]. The inclusion of winter rye in crop rotations is advantageous, as the ground cover it provides may reduce soil erosion [8], while simultaneously allowing a more sustainable management of the soil [9].

Pre-crops (previous crop) are used to add more organic matter to the soil while reducing nutrient leaching and stopping soil erosion by keeping the soil permanently covered with plants or their residues; pre-crops value is determined by long-term experiments [5]. Long-term experiments play a very important role in analyzing the stability of crop production, the changes in soil quality, technological progress, and the impact of environmental conditions on agriculture, as well as in assessing nutrient rotations. 

Soils are important sources of CO_2_ emission to the atmosphere. Introducing cover crop and conservation tillage is one of the strategies to improve soil organic carbon (SOC) and nitrogen (N) sequestration potentials with the potential to reduce the emission of greenhouse gases (GHG) [10]. 

Sosulski et al. [11] argued that in Central and Eastern European climates, soil respiration depends more on crop types and fertilization than on crop rotation systems. They conducted an experiment using a 90-year-old rye monoculture and crop rotations and found that under similar fertilization conditions, the amount of CO_2_ emission from the soil when growing cereals in monoculture and crop rotation was similar.

Many researchers argued that soil temperature is closely related to the release of CO_2_ into the environment: as the soil temperature increases, the release of CO_2_ into the environment becomes more intensive. Such effects are characterized by a feedback loop in the climate system [12,13,14].

Earthworms play an important role in the structure and fertility of soils, increasing the availability and release of C and N by stimulating the microbial mineralization of organic matter [15]. Earthworms can be used for organic waste management that promotes the recovery of nutrients from organic solid waste [16]. Earthworms can change soil structure and chemical and physical properties and alter the bioavailability and distribution of soil pollutants [15]. We hypothesize that crop rotation structure and monoculture have different effects on soil CO_2_ emission, as well as on earthworm populations and rye productivity.

The aim of this research was to investigate the effects of long-term crop rotation combinations on CO_2_ emission from the soil, earthworm populations, and the productivity of cereals.

## 2. Results and Discussion

### 2.1. Soil CO_2_ Emissions, Temperature, and Humidity

Soil is a major contributor of the active carbon (C) cycle in terrestrial ecosystems [17,18,19]. Daily, seasonal, and inter-seasonal dynamics of soil CO_2_ emissions are mainly driven by bioclimatic factors, including soil temperature and moisture [20,21,22].

In 2016, with green manure crop rotation, soil CO_2_ emissions in the winter rye crop stand changed little throughout the growing season (Figure 1). It can be assumed that the soil in which the green manure was incorporated gradually decomposed throughout the growing season. During the first 3 months, plant residues with more nitrogen decomposed most intensively. Holka and Bienkowski [23] also argued that nitrogen is particularly significant for GHG emissions, but cover crops and the large abandonment of crop residues in the field increase the sequestration of organic carbon and contribute to a significant reduction in carbon emissions from maize crops. The highest CO_2_ emissions during the first measurement (BBCH 73–75) were recorded in the field crop rotation with row crops in the winter rye crop stand after the 2nd year of use of perennial grasses. In intensive crop rotation, winter rye was grown after the 1st year perennial grasses, but more active CO_2_ emissions were recorded at the end of vegetation. The soil surface was covered all the time, as many catch crops (50% of the fields) were grown in intensive crop rotation, and at the end of winter rye vegetation, the conditions for microorganism activity were still favorable; however, relaxed nitrogen is required for rye that has started vegetation, which is why CO_2_ emissions were not intense. Nguyen and Kravchenko [24] investigated the intensity of CO_2_ emissions in sloping areas and found that in spring it depends on cover crops and environmental conditions. They also pointed out that winter rye emits less CO_2_ in spring compared to other crops (Figure 1). Other researchers pointed out that in Central and Eastern European climatic conditions, soil respiration is more dependent on the type of crop and fertilization than on crop rotation. They agreed with the aforementioned authors that an intense soil respiration is associated with plant yield and nitrogen uptake and also suggested that plants may increase autotrophic respiration. CO_2_ is released due to autotrophic respiration from photosynthesis, not due to the decomposition of soil organic matter [11].

Intensive respiration at the first measurement (BBCH 73–75) was determined for all crop rotations tested and for rye monoculture (Figure 2). The second measurement (BBCH 83–85) showed a 1.5-fold decrease in soil CO_2_ emissions, although soil moisture and temperature were higher than in the first measurement. The lowest soil CO_2_ emissions in crop rotations in winter rye crop stands were determined in three-course crop rotation. No significant differences between crop rotations and monoculture were found for winter rye crop throughout the growing season due to the large data dispersion between replications Sosulski et al. [11] also pointed out that under similar fertilization conditions, the distribution and the content of CO_2_ in cereal soils were similar with monocultures and crop rotation. The second measurement showed a 32.8% reduction in the CO_2_ emissions with the monoculture, which was not significantly different from the emissions from continuous bare fallow. As the moisture content increased, the soil lacked oxygen, and the high temperatures also inhibited microorganism activity. However, no correlation between the CO_2_ emissions and soil temperature and humidity was found. Research has shown that the daily, seasonal, and inter-seasonal dynamics of soil CO_2_ emissions are mainly driven by bioclimatic factors, including soil temperature and moisture [20,21,22].

Although there were no plants, continuous bare fallow promoted soil degradation—at the beginning of vegetation, CO_2_ emissions were present but lower compared to the ones during crop rotations. The incorporation of plant residues as well as straw into the soil increases the amount of carbon and does not reduce the CO_2_ emissions from the soil. In the three-course crop rotation, where rye was sown in fallow, the emissions were the lowest. Similar results were obtained by Canadian researchers who conducted long-term carbon conservation studies, comparing natural perennials with agricultural crops in the black ground zone. They found that in the 0–30 cm soil layer, wheat cultivation and straw legacy increased carbon stocks by an average of 0.6 t ha^−1^ per year, while with black fallow stocks, stocks carbon increased by only 0.23 t ha^−1^ [25]. Other researchers found that the emissions from fields cultivated without vegetation can be as high as 25–29% [26,27].

Similar results as those collected in 2016, were obtained in 2017 for winter rye. In the green manure crop rotation, the trend of soil CO_2_ emission intensity remained constant throughout the winter rye growing season. During the first and the second measurements, the highest CO_2_ emissions were found for the field crop rotation with row crops. The lowest CO_2_ emissions were found for a three-course crop rotation. Black fallow was kept in the three-course crop rotation before winter rye sowing, and the lack of organic matter was thought to contribute to the reduction of soil CO_2_ emissions after sowing the cereal. During vegetation, especially in the first half of summer, the intensity of CO_2_ increased, but not as much as during crop rotations. This result was reported by scientists growing maize for grain in three tillage systems (conventional, simplified, and no-tillage) to estimate greenhouse gas emissions and the costs associated with the production of maize. They concluded that carbon emissions in conventional, simplified, and no-tillage systems were 2347.4, 2353.4, and 1868.7 CO_2_ eq. ha^−1^, respectively.

Our long-term experiment with a traditional tillage system and black fallow in two crop rotations confirmed other scientists’ argument that it is disadvantageous from both an ecological and economic point of view. Cultivated soils are considered a source of CO_2_ emissions into the atmosphere [28,29]. It can be assumed that anthropogenic CO_2_ enters the atmosphere from the cultivated soil. Therefore, arable land is considered an important source of CO_2_ losses in the atmosphere [4]. Soil CO_2_ emissions integrate all components of soil CO_2_ production, including autotrophic and heterotrophic respiration. Greenhouse gas flux from soil is strongly influenced by soil conditions (e.g., organic matter content, bulk density, porosity, tillage), water management, fertilizer management, soil pH, pre-seasonal water status, and temperature [11,30,31,32,33,34,35,36,37,38,39]. In 2016, in the three-course crop rotation, the CO_2_ emissions increased towards the end of vegetation as soil moisture increased. In 2017, in intensive, green manure crop rotations and monoculture, the most intensive CO_2_ emissions were at the end of vegetation. It was observed that the soil moisture was higher during that period, although the temperature varied only slightly. However, in 2017, strong correlations were found between CO_2_ emissions and soil temperature (r = 0.89; *p* < 0.05) as well as soil moisture (r = 0.96; *p* < 0.05). With continuous bare fallow, it was also observed that with the increase of soil moisture, emissions increased in both years of the study, but not as intensively as with crop rotations.

### 2.2. Number and Mass of Earthworms in the Soil

The activity of earthworms and microorganisms causes endless changes on the physical, chemical, and biological characteristics of a substrate [40].

In 2016, after rye harvesting, the highest number of earthworms was found for the field crop rotation with row crops (Figure 3), which was significantly (1.8 to 2.1 times) higher compared to that found with the three-course and intensive crop rotations. For rye monoculture, the number of earthworms was similar to that for crops used in crop rotation. Retained rye straw is a nutrient source for soil fauna, no matter what cultivation technology is used. The most important biological component of the soil, earthworms, can be attracted by fresh root exudates and plant residues from additional sources of field biomass, such as cereals, and are then useful for decomposing stubble residues. Some researchers also pointed out that certain species of earthworms are found associated with most cereals and, under favorable conditions, their abundance does not decrease even though the cereals are harvested [41].

A significantly higher (from 1.9 to 4.5 times) earthworm mass was found for rye crop after field crop rotation with row crops compared to that found after three-course and intensive crop rotation and for rye monoculture. In the above-mentioned crop rotation, the earthworm mass was influenced by the pre-crop—a mixture of clover and timothy of the 2nd harvest year. Perennial grasses leave a sufficient amount of organic residues, and two years without disturbing the soil create favorable living conditions for earthworms.

In 2017, after winter rye harvesting, the same trends as in 2016 were observed for the number and the mass of earthworms (Figure 3). For the field crop rotation with row crops, the number of earthworms was significantly higher (from 2.2 to 11.1 times) compared to those found for other crop rotations and the rye monoculture. The lowest number of earthworms was found for the intensive crop rotation and was from 3.1 to 11.1 times lower than that for other crop rotations, although the pre-crop also consisted of perennial grasses though only grown for one year. It was observed that a higher soil moisture content was found in the field with row crops. The differences in temperature were small between the crop rotations. However, strong correlations between these indicators and the number of earthworms were obtained: r = 0.91, *p* < 0.05 and r = 0.99; *p* < 0.05, respectively. As shown in Figure 3, the same trend was observed for earthworm mass. A significantly higher (from 2.4 to 13.5 times) earthworm mass was found for rye crop of the field crop rotation with row crops compared to other crop rotations investigated. The lowest earthworm mass was recorded for the intensive crop rotation. In this crop rotation, catch crops were grown in 50% of the fields, and the topsoil was rather intensively disturbed, which might have had an adverse effect on the number and mass of earthworms. We compared the data between crop rotations during each investigation year. The activity of earthworms and microorganisms causes endless changes in the physical, chemical, and biological properties of the substrate [42]. This crop rotation was found to cause higher CO_2_ emissions from the soil, and a strong correlation was found between these indicators in the correlation analysis (r = 0.95; *p* < 0.05).

Earthworms can enhance micro-remediation and phytoremediation by increasing microbial activity and promoting plant growth [43,44].

### 2.3. Cereal Productivity

In 2016, the yield of winter rye ranged from 3.52 to 4.06 t ha^−1^ but did not differ significantly from that of 2017 (Figure 4). The yield of rye grown in monoculture was on average 21.5% higher compared to that obtained when rye was grown after other pre-crops. Such results were influenced by the extremely humid end of the growing season and the difficult harvesting period. Rye crops grown in crop rotations were lodged at the end of the vegetation period due to increased rainfall. Such a situation did not occur in the monoculture—rye did not lie down. In 2017, the highest winter rye yield was obtained when sowing after perennial grasses grown for two years, fertilized with cattle manure, and sown in cattle manure-treated black fallow. In monoculture, the yields of rye were significantly, on average by 22.5%, lower compared to those grown after perennial grasses and in black fallow but did not differ significantly compared to those grown after green manure. We compared the data was between crop rotations during each investigation year. Many researchers, including [11], suggest that growing agricultural crops in monocultures or in short-term sequences reduces their productivity. Thus, the results of our study only partially confirm this statement, as winter rye is one of the crops that does not require pre-crops. Winter rye productivity is influenced by the meteorological conditions of the year. Under favorable conditions, the productivity of monocultures does not decrease and is even comparable to that of crops grown in crop rotations. It should be noted that of the agricultural crops grown in Lithuania, maize and winter rye can best withstand continuous cultivation and growing in monocultures.

Winter rye was grown in the following sequences: grass (after perennial grasses), fallow (after fallow), green manure (after winter rape for green manure and monoculture, 49–51 years after rye). Although it is argued that in Lithuanian conditions higher yields are obtained when green manure crops are grown during the whole growing season [42], in 2016 with green manure crop rotation, ploughed-down winter rape, the amount of green mass did not have a significant effect on rye productivity. The yields in monoculture were by far the highest, but this was a consequence of the meteorological conditions. In 2017, winter rye yields were the best for field crop rotation with row crops, intensive, and three-course crop rotations. After fertilization of the monoculture and application of plant protection products, the yield of rye was equal to the yield of rye grown with green manure crop rotation.

## 3. Materials and Methods

### 3.1. Field Condition and Crop Rotations

A long-term stationary field experiment (Figure 5) was established in 1966 at Vytautas Magnus University Experimental Station in Lithuania (54°53′ N, 23°50′ E) and has been continued until now.

All crop rotations were arranged in time and space each year, three replications were applied. Each main plot was 18 m long by 9.60 m wide. The investigation was performed in 2016–2017 on winter rye (*Secale cereale* L.) “Matador”.

The study was conducted with 4 different crop rotations (Table 1): intensive, three-course, field rotation with row crops, green manure, as well as with a rye monoculture.

The soil of the experimental site was Endocalcari-Epihypogleyic Cambisol (sicco) (CMg-p-w-can) [45].

Granulometric composition—dusty loam on loam and clay. Average nutrient contents in the soil analyzed (2016): pH_KCl_—from 6.6 to 7.0; P_2_O_5_—from 131 to 206.7 mg kg^−1^; K_2_O—from 72.0 to 126.9 mg kg^−1^. Available K_2_O were analyzed by the Egner-Riehm-Domingo (AL) method [46]. C_org_—from 11.3 to 15.6 g kg^−1^ by a spectrophotometrical method.

During the experiment, the same arable tillage system was implemented, and plant protection products were used as needed.

### 3.2. Agronomic Management

Until 1990, all crop rotations were fertilized in such a way that during this period each crop rotation would receive the same amount of mineral fertilizers, and it was then fertilized at normal rates suitable for agricultural crops. In the experiments conducted in 2016 and 2017, tillage was carried out according to conventional cultivation technologies of winter rye (Table 2). After the first mowing, perennial grasses were disked in the field crop rotation with row crops and in intensive disked and incorporated as an organic fertilizer. In 2016 and 2017, early September, the soil was cultivated twice. The fertilizer of N_8_P_20_K_30_ 300 kg ha^−1^ was applied before the first cultivation each investigation year. Winter rye cv. “Matador” (180 kg ha^−1^) was sown in 18 September 2015 and 12 September 2016. The rye seed had not been treated. At the beginning of spring vegetation of winter rye, the plots were fertilized with 200 kg ha^−1^ of ammonium nitrate and additionally with 250 kg ha^−1^ after two weeks. Rye crop stands were protected from lodging and were sprayed with the growth regulators *Cycocel* 750 SL1 at 1.2 dm^3^ ha^−1^ (i.e., chlormequat chloride 750 g dm^3^) and *Stabilan* 750 SL (a.i. chlormequat chloride 750 g dm^3^). In spring, winter rye was sprayed with the herbicide *Arelon flussig* at 1.2 l ha^−1^ (i.e., isoproturon 50 gL^−1^) 2.0 l ha^−1^, 1.0 l ha^−1^, with fungicides *INPUT* 460 EC (i.e., prothioconazole 160 g l^−1^, spiroxsamine 300 gL^−1^) 1.0 l ha^−1^, and with *Fandango* (i.e., prothioconazole 100 gL^−1^, fluoxastrobin 100 gL^−1^) 1.0 l ha^−1^. Winter rye was harvested in 9 August 2016 and 3 August 2017.

### 3.3. Meteorological Conditions

In the spring of 2016, the conditions at the beginning of the vegetative growth of winter cereals were favorable. The air temperature was 1.6 °C higher than the long-term average. The air temperature in September 2016 was 1.3 °C higher than the long-term average. The autumn of 2016 was particularly warm, with positive temperatures prevailing until mid-December, and the average daily temperatures in September–December were 2.7 °C higher than the long-term average. Winter rye started tillering when preparing for winter. More significant negative temperatures were only recorded in January 2017 (−3.67 °C), but severe frosts did not arrive, and there was no snow cover. In March, the average temperature was 4.4 °C higher than the long-term average. Average annual temperatures and precipitation in 2015–2017 are presented in Table 3 and Table 4.

In July 2016, the rainfall was twice as much as usual, and the autumn was cool and very rainy. The temperature in October was 1.5 °C lower than the long-term average, while the amount of rainfall exceeded the long-term average by 81.9 mm. In November, precipitation was 20.7 mm higher than usual. In winter, precipitation was close to the long-term average; however, the soil was very wet due to excess moisture in autumn. In 2017, precipitation was significantly larger, particularly in April (73.7 mm) when it exceeded the long-term average by 35.3 mm. Plants suffered from excess moisture.

Precipitation varied greatly from month to month. At the beginning of vegetation, it was excessively humid, but during other vegetative stages, it was dry, especially in May, July, and August.

### 3.4. Methods and Analysis

Soil CO_2_ emissions were measured using an Infra-Red Gas Analyzer to measure the soil CO_2_ efflux (μmol m^−2^ s^−1^). The portable, automated soil gas flux system LI-8100A with an 8100-103 chamber and the analyzer LI-8100A (LI-COR Inc.) was used. In spring, 20 cm-diameter rings were installed in the soil in each plot, and there were no growing plants. Two days before the measurements, all grew plants were removed. Three measurements were made in each plot. CO_2_ efflux was determined three times per growing season, at the same time of the day (from 10 a.m. to 3 p.m.) and at designated locations in the field. Soil moisture was measured with the sensor LI-8100-204 (LI-COR Inc.), and soil temperature with the sensor LI-8100-203 (LI-COR Inc.), included in the chamber control of the LI-8100A automated soil gas flux system (LI-COR Inc.). In Equation (1):(1)$$\frac{\partial C_{c}\left(t\right)}{\partial t}=A\left(C_{s}-C_{c}\left(t\right)\right)$$
*C_s_* is CO_2_ concentration (µmol/mol) in the soil surface layers, and *A* (s^−1^) is a rate constant that is proportional to CO_2_ conductance at the soil surface and to the surface-to-volume ratio of the chamber. If *A* and *C_s_* are constant, then integration with respect to time gives:(2)$$C_{c}\left(t\right)=C_{s}+\left(C_{c}\left(0\right)-C_{s}\right)e^{-At}$$

In the LI-8100 system, the chamber CO_2_ concentration *C_c_(t)* versus time data are fitted with an exponential function of the form given in Equation (2), yielding values for the parameters *A* and *C_s_*. Soil CO_2_ flux is then obtained by calculating the initial slope (*∂C_c_*(*t*))/*∂t* from Equation (1) at time zero when the chamber touches down and *C_c_*(0) = ambient. A complete description of the equations used in the LI-8100 system, including details of dilution corrections due to water vapor, is given in the LI-8100 Instruction Manual (2004). Soil temperature and humidity measurements were made during fixation by soil CO_2_ emission. These measurements were made at the beginning, middle, and end of vegetation.

The number and the mass of earthworms in the soil were determined after harvesting by the chemical repellent method [47]. In each plot, three frames (50 × 50 cm) were hammered into the soil at a depth of 10 cm. A formalin solution with a concentration of 0.55% was applied twice every 15 min. The emerged earthworms were collected, counted, and weighed. The amount and mass of earthworms were measured after yield harvesting.

A special plot harvester (Wintersteiger AG, Ried im Innkreis, Austria) was used for pre-crop harvesting. Cereal grain yield was adjusted to 14% moisture 100% grain mass purity.

### 3.5. Statistical Analysis of the Experimental Data

The research data were processed by the method of analysis of variance using the computer program STATISTICA SIX SIGMA10 [48]. The Kruskal–Wallis test (*p* < 0.05) was applied to the data that did not show a normal distribution (e.g., soil CO_2_ emissions). The research data (number of earthworms, their mass, rye yield) were statistically evaluated by one-way analysis of variance (ANOVA) of quantitative traits; the LSD test was also applied. Data regarding the number and mass of earthworms that did not conform to the normal distribution law were transformed using the Log (X) function before statistical evaluation. The method of correlation regression analysis was applied to evaluate the causality of the studied traits. We used the program STAT ENG from the package ANOVA [49,50].

## 4. Conclusions

This study confirmed our hypothesis that long-term crop rotation has a positive effect not only on the yield of winter rye but also on soil CO_2_ emissions as well as earthworm population. In the field with row crops, plant residues of perennial grasses increased the intensity of CO_2_ emissions especially in the early spring vegetation of winter rye. Growing rye in black fallow tended to reduce the emission intensities, but frequent storage of fallow is not economically and ecologically beneficial. Plant residues of perennial grasses and two years of uncultivated soil in the field with row crops substantially increased the earthworm population. The earthworm population was negatively affected by intensive crop rotation with catch crops and a frequent storage of fallow in the three-course crop rotation, which was associated with intensive tillage. Correlation between the soil CO_2_ emissions and the mass of earthworms was found to be r = 0.95; *p* < 0.05. Rye productivity was affected by annual meteorological conditions and the pre-crop. In 2016, rye yields decreased due to environmental factors. In the field with row crops, in which rye was grown after the pre-crop of perennial grasses grown for the second year, the yield of 2017 was the highest—it was 27.0% higher compared to that of monoculture and of rye grown after green manure. Pre-crop of perennial grasses used in the second year increased CO_2_ emissions and the number and mass of earthworms, but no correlation was found between these indicators and the yields.

## Figures and Tables

**Figure 1 plants-11-00431-f001:**
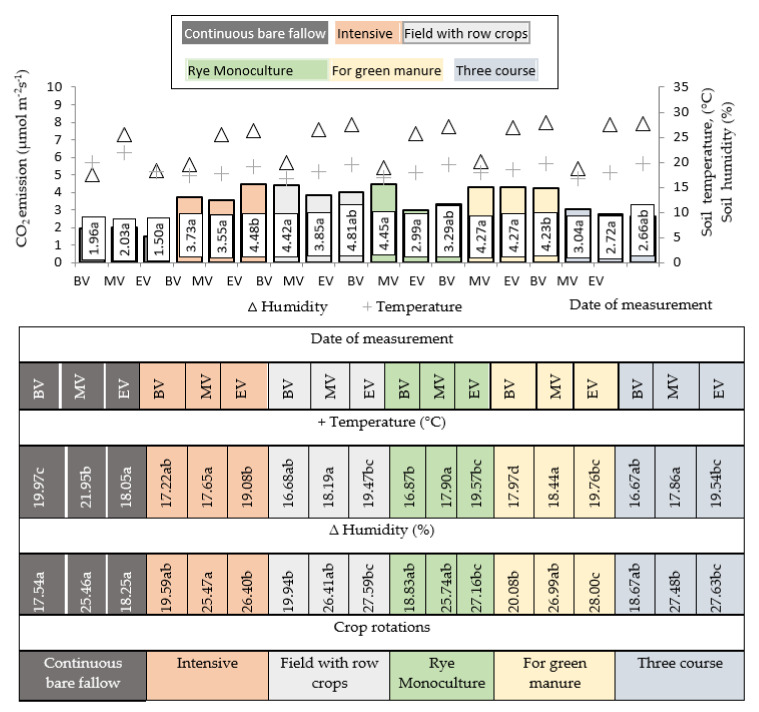
Soil humidity (%), temperature (°C), and CO_2_ emission (µmol m^−2^s^−1^) for winter rye crop in 2016. **Notes.**
^a–d^ different letters indicate significant differences between the treatments (*p* ≥ 0.05); Date of measurement: BV—the beginning; MV—the middle; EV—the end of rye vegetation.

**Figure 2 plants-11-00431-f002:**
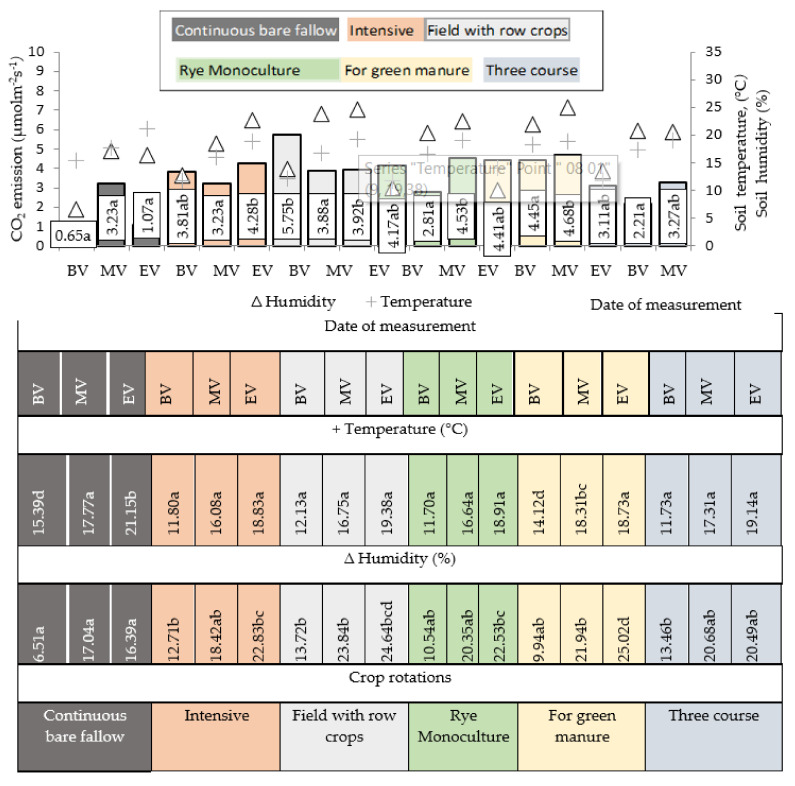
Soil humidity (%), temperature (°C), and CO_2_ emissions (µmol m^−2^s^−1^) for winter rye crop in 2017. **Notes.**
^a–d^ different letters indicate significant differences between the treatments (*p* ≥ 0.05); Date of measurement: BV—the beginning; MV—the middle; EV—the end of rye vegetation.

**Figure 3 plants-11-00431-f003:**
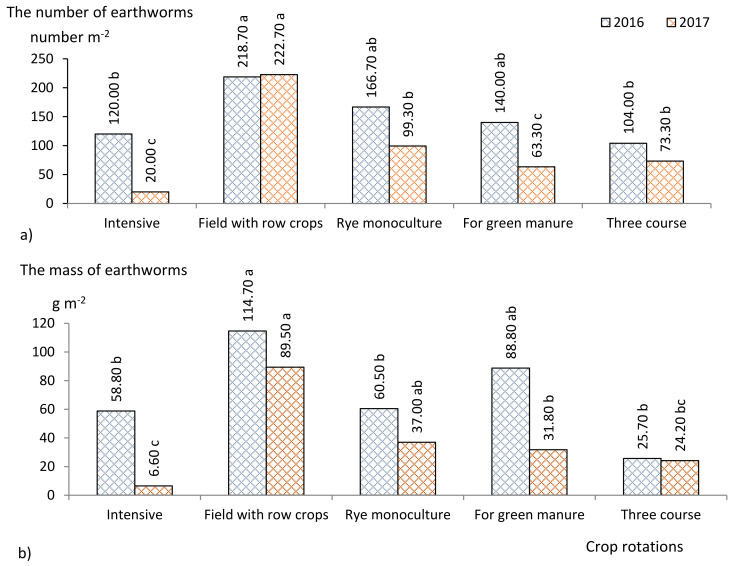
(**a**) number (number m^−2^) and (**b**) mass (g m^−2^) of earthworms associated with winter rye crop in 2016 and 2017. **Notes.**
^a–d^ different letters indicate significant differences between the treatments (*p* ≤ 0.05).

**Figure 4 plants-11-00431-f004:**
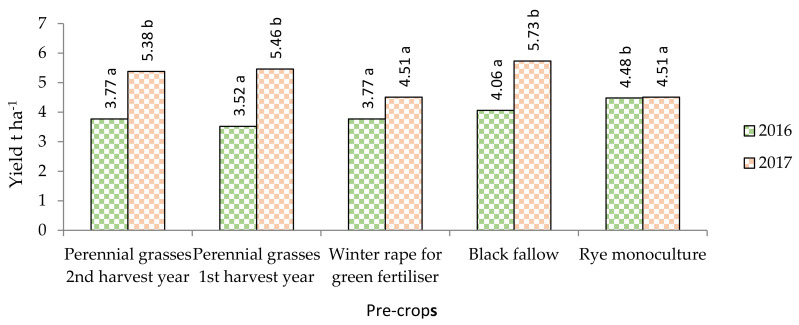
Winter rye grain yield (t ha^−1^) average after various pre-crops in different crop rotations in 2016 and 2017. **Notes.**
^a–d^ different letters indicate significant differences between the treatments (*p* ≤ 0.05).

**Figure 5 plants-11-00431-f005:**
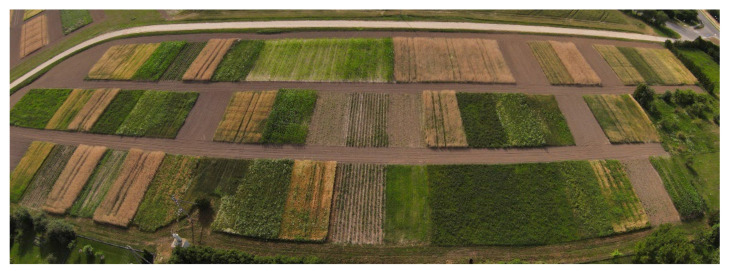
Image of crop rotations at Vytautas Magnus University Experimental Station (54°53′ N + 23°50′ E).

**Table 1 plants-11-00431-t001:** Crop rotation sequences.

Crop Rotation	Crop Rotation Components
Intensive	(1) Vetch-oat (*Vicia sativa* L. *+ Avena sativa* L.) mixture for fodder + undersow; (2) Perennial grasses (*Trifolium pratense* L.+*Phleum pratense* L.) (first year); (3) Winter rye (*Secale cereale* L.) and, after an intermediate crop, winter rape (*Brassica napus* L.); (4) Potatoes (*Solanum tuberosum* L.) and, after an intermediate crop, winter rye (*Secale cereale* L.) for fodder; (5) Corn (*Zea mays* L.); (6) Spring barley (*Hordeum vulgare* L.) and, after an intermediate crop, oil radishes (*Raphanus sativus* L.).
Field rotation with row crops	(1) Winter wheat (*Triticum aestivum* L.) + undersow; (2) Perennial grasses grasses (*Trifolium pratense* L*. + Phleum pratense* L.) (first year); (3) Perennial grasses grasses (*Trifolium pratense* L*. + Phleum pratense* L.) (second year); (4) Winter rye (*Secale cereale* L.); (5) Sugar beet (*Beta vulgaris* L.); (6) Spring barley (*Hordeum vulgare* L.); (7) Oat (*Avena sativa* L.); (8) Black fallow.
Rye monoculture	(1) Winter rye (*Secale cereale* L.).
Green manure	(1) Lupines (*Lupinus angustifolius* L.) for green manure; (2) Winter rye (*Secale cereale* L.*)*; (3) Winter rape (*Brassica napus* L*.)* for green manure; (4) Winter rye (*Secale cereale* L.*)*; (5) Potatoes (*Solanum tuberosum* L.*)*; (6) Spring barley (*Hordeum vulgare* L.).
Three-course	(1) Black fallow; (2) Winter rye (*Secale cereale* L.); (3) Oat (*Avena sativa* L.).

**Table 2 plants-11-00431-t002:** Sources of organic matter in crop rotations.

Crop Rotations	CROPS	SOURCE of Organic Matter			
		MANURE (55 t ha^−1^)	STRAW	Green Manure	Perennial Grasses
Intensive	Winter rye	+	+	+	+
Field with row crops	Winter rye	+	+		+
Rye monoculture	Winter rye		+		
For green manure	Winter rye		+	+	
Three-course	Winter rye		+		

**Table 3 plants-11-00431-t003:** Average temperature (°C) and the sum of the active temperatures (SAT) during the winter rye growing season (September–August) in 2015–2017.

Year/Month	09	10	11	12	01	02	03	04	05	06	07	08	SAT
2015–2016	14.3	6.2	4.9	2.6	−7.1	0.6	2.1	7.4	15.7	17.2	17.9	16.9	2544.7
2016–2017	13.5	5.3	1.2	1.2	−3.7	−1.5	3.7	5.6	12.9	13.4	16.8	17.5	2331.5
Long-term average 1974–2018	12.6	6.8	2.8	−2.8	−3.7	−4.7	0.3	6.9	13.2	16.1	18.7	17.3	-

SAT = sum of active temperatures (≥10 °C), Kaunas Meteorological Station.

**Table 4 plants-11-00431-t004:** Precipitation (mm) during the winter rye growing season (September-August) in 2015–2017.

Year/Month	09	10	11	12	01	02	03	04	05	06	07	08	Sum
2015–2016	56.6	18.2	95.6	61.3	41.6	68.4	47.2	41.2	36.4	83.9	162.9	114.9	828.2
2016–2017	22.5	101.5	66.8	56.5	18.4	31.3	53.1	73.7	10.5	80.2	79.6	5.5	599.6
Long-term average 1974–2018	60.0	51.0	51.0	41.9	38.1	35.1	37.2	41.3	61.7	76.9	96.6	88.9	679.7

Kaunas Meteorological Station.

## Data Availability

Not applicable.

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
