# Peer review of "The Effect of Monoculture, Crop Rotation Combinations, and Continuous Bare Fallow on Soil CO2 Emissions, Earthworms, and Productivity of Winter Rye after a 50-Year Period"

_plants, 2022, doi:10.3390/plants11030431_

Round 1
Reviewer 1 Report
The manuscript presents interesting research results, but their value is lowered by the fact that they only come from a two-year period. The manuscript requires some corrections, mainly in the Materials and methods section.
Comments:
I suggest that you consider changing the title so that it does not enumerate the features assessed in the experiment
Line 30, surface1?
Line 56, correct Sosulski et al. [11]
Line 79, please include a figure with the location of the test site
Line 82, unclear whether the same variety of winter rye was cultivated since 1996 or in 2016-2017?
Line 90, please complete the characteristics of soil conditions with the content of soil organic carbon and micronutrients?
Line 91, pH in KCl or H2O?
Line 91, please provide P and K content in mg P and K per kg soil, not in mg P2O5 and K2O
Line 104, has the rye seed been treated?
Line 104, please provide the date of sowing and harvesting of rye
Line 106, please provide the total number of doses N, P and K
Line 107 etc. correct 1.2 dm3 ha-1
Line 107 etc. correct 750 g dm-3
Table 2, I suggest 55 t ha-1, not 55 Mg ha-1 (look Figure 4)
Line 126, too large space in front of the table
Line 126, 136, source below the table
Line 181, 189, please provide full details of the manufacturer of the statistical software
Line 279, correct [11, 37–46].
Line 339, lodging - why green font?
References, I propose to delete the oldest publications, older than 2010 (2, 7, 8, 15, 17, 18, 20, 23, 28, 32, 33, 37, 43, 44, 45, 46, 49,
References, please adapt to the requirements of the journal.
Author Response
[2022-01-31]
Dear Reviewer:
We wish to submit an original research article for publication in Plant. We have made significant improvements to this article, titled “The effect of monoculture, crop rotation combinations, and continuous bare fallow on soil CO2 emissions, earthworms, and productivity in winter rye after a 50-year period”.
We would like to sincerely thank You for your insightful comments and recommendations that allowed us to fundamentally correct the article.
The manuscript presents interesting research results, but their value is lowered by the fact that they only come from a two-year period. The manuscript requires some corrections, mainly in the Materials and methods section.
Comments:
- I suggest that you consider changing the title so that it does not enumerate the features assessed in the experiment
We thank the Reviewer for the comment. We think that the title is informative and we prefer not to change the title.
- Line 30, surface1?
It was corrected.
- Line 56, correct Sosulski et al. [11]
It was corrected.
- Line 79, please include a figure with the location of the test site
Added figure with the location of the test site.
- Line 82, unclear whether the same variety of winter rye was cultivated since 1996 or in 2016-2017?
It was corrected.
- Line 90, please complete the characteristics of soil conditions with the content of soil organic carbon and micronutrients?
The soil organic carbon was added.
- Line 91, pH in KCl or H2O?
It was corrected.
- Line 91, please provide P and K content in mg P and K per kg soil, not in mg P2O5 and K2O
We added the method how were established P2O5 and K2O. We can not to provide in P and K content in mg.
- Line 104, has the rye seed been treated?
We thank the Reviewer for the comment.
- Line 104, please provide the date of sowing and harvesting of rye
We thank the Reviewer for the comment. Sowing and harvesting dates were added.
- Line 106, please provide the total number of doses N, P and K
We thank the Reviewer for the comment
- Line 107 etc. correct 1.2 dm3 ha-1
It was corrected.
- Line 107 etc. correct 750 g dm-3
It was corrected.
- Table 2, I suggest 55 t ha-1, not 55 Mg ha-1 (look Figure 4)
It was corrected.
- Line 126, too large space in front of the table
It was corrected.
- Line 126, 136, source below the table
It was corrected.
- Line 181, 189, please provide full details of the manufacturer of the statistical software
We thank the Reviewer for the comment.
- Line 279, correct [11, 37–46].
It was corrected.
- Line 339, lodging - why green font?
It was corrected.
- References, I propose to delete the oldest publications, older than 2010 (2, 7, 8, 15, 17, 18, 20, 23, 28, 32, 33, 37, 43, 44, 45, 46, 49,
We thank the Reviewer for the comment, but we think that used investigations are important and these days and their investigated time does not change the importance of the research.
References, please adapt to the requirements of the journal.
It was corrected.
We reviewed and clarified the manuscript. Thank you for your consideration.
Sincerely,
Lina Skinulienė
Vytautas Magnus University, K. Donelaičio str. 58, 44248 Kaunas, Lithuania
[+37067412525]
lina.skinuliene@vdu.lt
Reviewer 2 Report
Dear authors
The paper is excellent, great ideas and good measurements and experiments
Please, see attached... your contribution is excellent

Author Response
[2022-01-31]
Dear Reviewer:
We wish to submit an original research article for publication in Plant. We have made significant improvements to this article, titled “The effect of monoculture, crop rotation combinations, and continuous bare fallow on soil CO2 emissions, earthworms, and productivity in winter rye after a 50-year period”.
We've taken the comments and made the changes.
We would like to sincerely thank You for your comments and recommendations that allowed us to correct the article.
We reviewed and clarified the manuscript. Thank you for your consideration.
Sincerely,
Lina Skinulienė
Vytautas Magnus University, K. Donelaičio str. 58, 44248 Kaunas, Lithuania
[+37067412525]
lina.skinuliene@vdu.lt
Reviewer 3 Report
The manuscript entitled “The effect of monoculture, crop rotation combinations, and continuous bare fallow on soil CO2 emissions, earthworms, and productivity in winter rye after a 50-year period " is a successful illustration of a holistic analysis of factors determining soil respiration in the production and environmental dimensions. The authors examined the amount of CO2 released from the soil under the cereal that is important in this part of Europe - winter rye - perfectly adapted to soil and climatic conditions of Lithuania. The manuscript, due to the results of research from many years of field experiments, provides important information on soil respiration in various crop rotation systems, which is essential for meeting the basic expectations of the European Green Deal (if agriculture is to take an active part in reducing GHG emissions to the zero-emission level , or if it will be charged with emission fees). The authors focused not only on assessing the impact of plant alternation / monoculture on soil respiration. The research results indicate that the issue of soil biological activity is very often overlooked in such studies. This gives added value to these studies. I rate the work very highly, and the comments are marginal:
Line 91 - please provide the type of soil test (e.g. Mehlich-3, Egner (DL)?)
316 double dots
339 - greenish font?
Author Response

(The authors gave the same response as above.)
